

# A novel, balloon-borne UV/visible spectrometer for direct sun measurements of stratospheric bromine

Karolin Voss[1], Philip Holzbeck[1,2], Klaus Pfeilsticker[1], Ralph Kleinschek[1], Gerald Wetzel[3], Blanca Fuentes Andrade[1,4], Michael Höpfner[3], Jörn Ungermann[5], Björn-Martin Sinnhuber[3], and André Butz[1,6,7]

[1]Institute of Environmental Physics, Heidelberg University, Germany
[2]now at Atmospheric Chemistry Department, Max Planck Institute for Chemistry, Mainz, Germany
[3]Institute of Meteorology and Climate Research, Karlsruhe Institute of Technology (KIT), Karlsruhe, Germany
[4]now at Institute of Environmental Physics, University of Bremen, Germany
[5]Institute of Energy and Climate Research, Forschungszentrum Jülich (FzJ), Jülich, Germany
[6]Heidelberg Center for the Environment, Heidelberg University, Germany
[7]Interdisciplinary Center for Scientific Computing, Heidelberg University, Germany

**Correspondence:** Karolin Voss (karolin.voss@uni-heidelberg.de)

**Abstract.** We report on a novel, medium weight ($\sim 25\,\mathrm{kg}$) optical spectrometer coupled to an automated sun tracker ($\sim 12\,\mathrm{kg}$) for direct sun observations from azimuth controlled balloon platforms. It is designed to measure a suite of UV/vis absorbing gases ($O_3$, $NO_2$, BrO, OClO, HONO and IO) relevant in the context of stratospheric ozone depletion using the DOAS method. Here, we describe the design and major features of the instrument. Further, the instrument's performance during two stratospheric deployments from Esrange/Kiruna (Sweden) on 21 August 2021 and from Timmins (Ontario, Canada) on 23 August 2022 are discussed along with first results concerning inferred mixing ratios of BrO above balloon float altitude. Using a photochemical correction for the partitioning of stratospheric bromine ($[\mathrm{BrO}]/[\mathrm{Br_y}]$) obtained by chemical transport simulations, the inferred total stratospheric bromine load $[\mathrm{Br_y}]$ amounts to $(17.5 \pm 2.2)\,\mathrm{ppt}$ (pure statistical error amounts to $1.5\,\mathrm{ppt}$) in $(5.5 \pm 1.0)\,\mathrm{yrs}$ old air, resulting in a stratospheric entry early $2017 \pm 1\,\mathrm{yr}$, the latter being inferred from simultaneous measurements of $N_2O$ by the GLORIA mid-IR instrument.

## 1 Introduction

Although much less abundant than chlorine, bromine contributes about 1/3 to the columnar loss in stratospheric ozone since it has an ozone depletion potential as large as 60 to 75 (Ko et al. (2003), Sinnhuber et al. (2009), Engel et al. (2018), Koenig et al. (2020), Klobas et al. (2020), Laube et al. (2022), and others). Sources of stratospheric bromine (in 2021/2022) include man-made brominated organic species ($\sim 1.5\,\mathrm{ppt}$ from $CH_3Br$ and $\sim 7.1\,\mathrm{ppt}$ from the halons), naturally emitted $CH_3Br$ ($\sim 5.5\,\mathrm{ppt}$) and so called brominated very short-lived substances (VSLS) and their organic decay products ($\sim 5\,\mathrm{ppt}$). The total





of these sources reached a maximum of about 22.1 ppt around the turn of the century but declined ever since to $(19.2\pm1.2)$ ppt in 2017/2018, mostly due to the emission reduction regulations agreed on in the Montreal Protocol (Rotermund et al., 2021; Laube et al., 2022). Under a changing climate, future stratospheric bromine abundances could increase again, mostly due to stronger marine emissions and the more efficient delivery of brominated VSLS to the stratosphere, but the extent of these changes remains speculative (Falk et al., 2017).

However, it is not fully clear what the notion 'delivery to the stratosphere' really means for the stratospheric budget of bromine, since bromine enters the stratosphere via (a) the tropical tropopause layer, (b) transport within the lower branch of the Brewer Dobson circulation, and (c) extra-tropical stratosphere-troposphere exchange. At the same time, bromine may undergo uncertain transformation processes of the gaseous species into particulate form (and vice versa) and hence may become subject to heterogeneous removal (Sinnhuber and Folkins (2006), Werner et al. (2017), Rotermund et al. (2021), and others). Therefore, bromine amounts in the lowermost stratosphere may (slightly) differ from its amounts in the middle stratosphere. Comparing the bromine transported through the tropical tropopause layer (e.g. Werner et al. (2017), Wales et al. (2018)) and the bromine found in the tropical middle stratosphere or in the descending branch of the Brewer Dobson circulation (e.g. Dorf et al. (2008), Rotermund et al. (2021)) hints at differences of $1-2$ ppt. In consequence to further quantify all these processes, it appears desirable to precisely measure and specify all forms of bromine in all sub-domains of the stratosphere.

Traditionally, stratospheric bromine abundances have often been inferred from measured bromine monoxide (BrO) or bromine nitrate ($BrONO_2$) concentrations (employing a suitable correction for the bromine partitioning) by using in-situ and remote sensing instrumentation, which were deployed on crewed and uncrewed aircraft, high flying balloons, and satellites (e.g. Brune et al. (1989), Toohey et al. (1990), Harder et al. (1998, 2000), Pundt et al. (2002), Sioris et al. (2006), Theys et al. (2009), Höpfner et al. (2009, 2021), Rozanov et al. (2011), Liao et al. (2012), Stachnik et al. (2013), Volkamer et al. (2015), Werner et al. (2017), Wetzel et al. (2017), Rotermund et al. (2021), and others). Among the various employed techniques (e.g. Resonance Fluorescence, Chemical Ionisation Mass Spectrometry, Middle Infrared and Microwave Spectroscopy, and Differential Optical Absorption Spectroscopy), the Differential Optical Absorption Spectroscopy (DOAS) for direct sunlight (solar occultation) and atmospheric skylight measurements has most widely been used for BrO observations in the lower and middle stratosphere, due to its excellent selectivity, sensitivity, and repeatability (Harder et al. (1998, 2000), Pundt et al. (2002)). In particular, the repeatability of these DOAS observations helped establish the trend in total stratospheric bromine in the past three decades, which is necessary for the verification of the regulations of the Montreal protocol (Dorf et al. (2006b), WMO (2022)).

From 1996 until 2012, we regularly deployed a DOAS instrument for solar occultation measurements of halogen oxides (BrO, chlorine dioxide (OClO), and iodine oxide (IO)) in the stratosphere on the azimuth-stabilised LPMA/DOAS (Laboratoire de Physique Moléculaire et Applications) balloon gondola (Camy-Peyret et al. (1993, 1995, 1999), Ferlemann et al. (1998, 2000)). However, due to a change in ballooning technology and operations by CNES (Centre National d'Étude Spatiale) in the mid 2010s, the previously used LPMA/DOAS balloon gondola and DOAS solar occultation instrument were decommissioned. Here, we present a newly developed, compact DOAS spectrometer system coupled to an active solar tracker for balloon-borne solar occultation measurements of mid-stratospheric BrO and possibly, to be addressed by future studies,





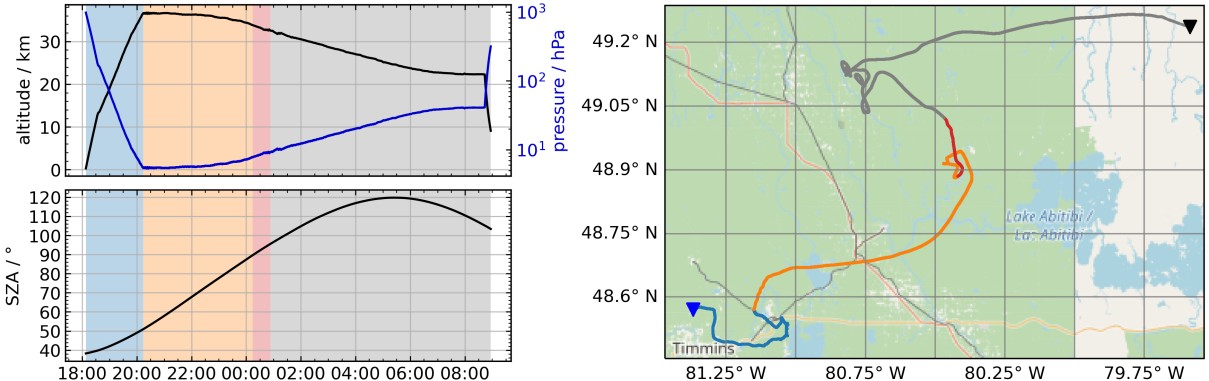

**Figure 1.** Flight profile (left) of the second deployment from Timmins, Canada, (23 Aug 2022) showing balloon altitude, air pressure (upper panel), and SZA over time in UTC (lower panel). The shadings indicate the distinct phases of the flight: balloon ascent (blue), balloon float with SZA < 90° (orange), solar occultation (red) and night (grey). The balloon's trajectory (right) is marked by the same flight phase colours. The blue triangle marks the launching site and the black triangle marks the landing site of the balloon. Due to a leakage of the helium balloon it descended slowly after reaching the maximum altitude (36 km) around 20:00 UTC. Map from Open Street Map.

other UV/visible absorbing gases such as IO and OClO. We report on the instrument setup, the demonstration deployments and the performance of the new system for BrO concentration measurements and on the derived total inorganic bromine ($[Br_y]$).

The paper is structured as follows. Section 2 reports on the two deployments to date, Sect. 3 describes the design and technical features of the spectrometer system and the sun-tracker as well as its performance, and Sect. 4 discusses the different methods used to infer total bromine from measured BrO. Section 5 reports on the stratospheric abundances inferred from the second deployment and discusses the obtained results and Sect. 6 concludes the study.

## 2   Deployments

To date, the new instrument has been deployed twice onboard azimuth-stabilised stratospheric balloon gondolas. Here, we will mainly report on the second deployment, which took place from Timmins, Ontario, Canada (48.57° N, 81.37° W) on 23 August 2022 in the early afternoon (launch at 18:08 UTC, 14:08 local time). The flight profile and trajectory of this balloon flight are shown in Fig. 1. We collected good quality measurements under afternoon and sunset conditions (solar zenith angles (SZA) between 55° and 96°) from a balloon ceiling altitude between 32 km and 36.5 km. Measurements during the ascent of

the balloon into the stratosphere were not possible since the azimuth rotation of the gondola could not be stabilised due to high wind shear. Other remote sensing payloads on this flight were GLORIA (Gimballed Limb Observer for Radiance Imaging of the Atmosphere by Karlsruhe Institute of Technology (KIT) and Forschungszentrum Jülich (FzJ), Riese et al. (2014); Friedl-Vallon et al. (2014); Höpfner et al. (2022)) and FIRMOS (Far-Infrared Radiation Mobile Observation System by National Institute of Optics (CNR-INO), Belotti et al. (2023); Palchetti et al. (2021)).





One year before, on 21 August 2021, the first deployment of our instrument took place from Esrange near Kiruna, Sweden (67.89° N, 21.08° E) in the afternoon (launch at 15:08 UTC, 17:08 local time). This flight (total flight time around 18 h) allowed us to record spectra during sunset (SZA between 85° and 95°) and subsequent sunrise (SZA between 94.5° and 61°) from balloon altitudes above 32 km. Measurements during the ascent of the balloon into the stratosphere were impossible due to both technical problems with our instrument and difficulties with the azimuth-stabilisation of the gondola. GLORIA and

ALI (Aerosol Limb Imager by the Institute of Space and Atmospheric Studies at the University of Saskatchewan, Elash et al. (2016)) were other scientific payloads on the same flight. While we report on the performance of the solar tracker for the first flight, spectroscopic artefacts (etalon structures) hinder us from carrying out and reporting on the spectral analysis. We were able to reduce these artefacts for the second flight from Timmins to the extent that we can correct from them (see Sect. 4.1). Thus, our spectroscopic results mainly derive from the second flight.

## 3  Instrumentation

The new balloon-borne DOAS instrument is designed to measure UV/visible absorbing trace gases such as $O_3$, $NO_2$, BrO, possibly IO, OClO and HONO under stratospheric pressure and temperature conditions. The instrument is of medium weight (total mass $< 40$ kg) and has a low power consumption ($< 100$ W), making it a suitable secondary payload on stratospheric balloon gondolas. The balloon-borne DOAS instrument, depicted in Fig. 2, consists of two major parts: an active solar tracking

unit capturing direct sunlight (described in Sect. 3.1) and the spectrometer unit containing two grating spectrometers (described in Sect. 3.2), one sensitive to the UV and the other one to the visible (vis) spectral range. Two telescopes within the solar tracker feed the direct sunlight into the spectrometers via two glass fibres of 3 m length each. Each unit is equipped with an embedded computer (Fitlet2-CE3959-P36) and an Arduino-based housekeeping unit logging temperature data. All electronics threatened to overheat are thermally connected to a radiation shield (hereinafter called radiator) attached to the respective unit

and facing the cold sky to provide radiative cooling during stratospheric balloon flights. In addition, baffles shield the radiators from direct sun radiation. The embedded computers can be operated remotely from the ground via telecommand/telemetry connections provided by the host gondola. Each unit is connected to a GPS receiver to synchronise its clock. Thus, except for the connection via the glass fibres, the spectrometer and solar tracker units are two stand-alone units that can be placed in different parts of the host gondola, e.g. the solar tracker can be installed on the outside fuselage of the gondola to enable

direct sun viewing while the spectrometer unit can be placed somewhere inside the structure, with the side-constraint that the radiators need to face the cold background sky.

### 3.1  Solar tracker unit

The concept of the stand-alone solar tracker is based on the Camtracker setup developed by Gisi et al. (2011) and used routinely for ground-based direct sun observations by our group and many others (e.g. Frey et al. (2019)).

Two plane ellipse-shaped mirrors are assembled in an alt-azimuth mount on two motorised stages (STANDA 8MR151-1 and STANDA 8MR190-2-28) whose lubricant has been replaced by a Teflon-based lubricant (Fluroxon GV2S, specified from



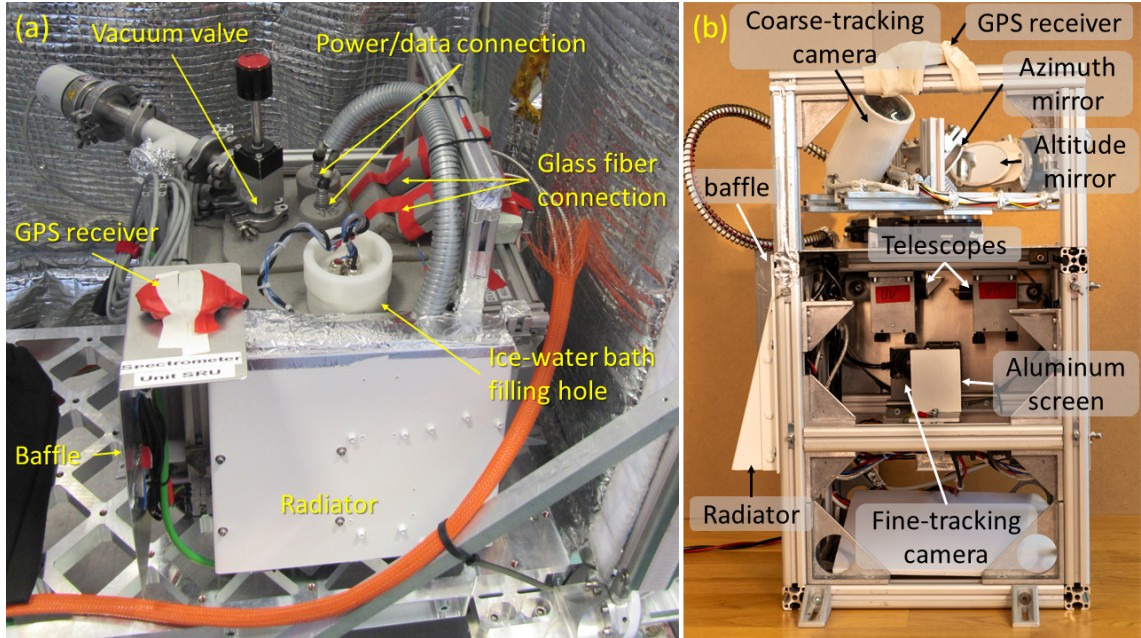

**Figure 2.** Photographs with labelling of the important components of the spectrometer (a) and solar tracker unit (b). Note that the vacuum valve and tubing are unmounted during the actual balloon flight since vacuum pumping is only performed at ground prior to the flight. Also, the open side of the solar tracker is closed by an aluminum plate during the flight.

$-60\,°\text{C}$ to $100\,°\text{C}$) to avoid clogging under stratospheric conditions. Additionally, each stage motor is kept warm by a self regulating heater (DBK HP05-104, each $\leq 15\,\text{W}$). The plane mirrors reflect the parallel beam of sunlight into the tracker's body, where it is deflected by a $45°$ folding mirror and focused by a lens ($2.54\,\text{cm}$ ($1\,\text{inch}$) diameter, focal length $f = 150\,\text{mm}$)
onto a sand-blasted aluminum screen positioned at a distance of one focal length from the lens. The image of the sun on the screen is approximately $1.4\,\text{mm}$ in diameter, which equals 64 pixels in the picture of the so-called fine-tracking camera inside the tracker's body. The tracker unit measures $0.4\,\text{m} \times 0.4\,\text{m} \times 0.5\,\text{m}$, weights around $12\,\text{kg}$, and consumes around $60\,\text{W}$ at peak computational and heating power. The tracker is depicted in Fig. 2(b).

     The tracking operates in two steps: first, a coarse tracking fish-eye lens camera with a field of view of $185°$ (IDS UI-3280CP
Rev.2 with Fujinon FE185C057HA-1 f-theta objective lens) mounted at a zenith angle of $45°$ finds the rough position of the sun in the sky. The tracker rotates the altitude and azimuth mirrors accordingly, such that the sun's image is visible by the fine-tracking camera that observes a target area of roughly $21\,\text{mm}$ diameter on the sand-blasted aluminum screen. Then, the faster fine-tracking camera (IDS UI-3140CP Rev.2) takes over and slightly adjusts the mirror positions to centre the sun's disc on a predefined centre position using a PID control loop with a frequency of $50\,\text{Hz}$. The target precision of the tracking is $\approx \frac{1}{10}$
of the sun's diameter ($0.05°$ full angle). In principle, sun tracking is possible down to a solar zenith angle (SZA) of $96°$ and in an azimuth segment of $360°$. During the two deployments reported here, the tracker was mounted on the top of the balloon gondola structure on the side facing the sun to ensure an unobscured view of the sun throughout the flight. When mounted on





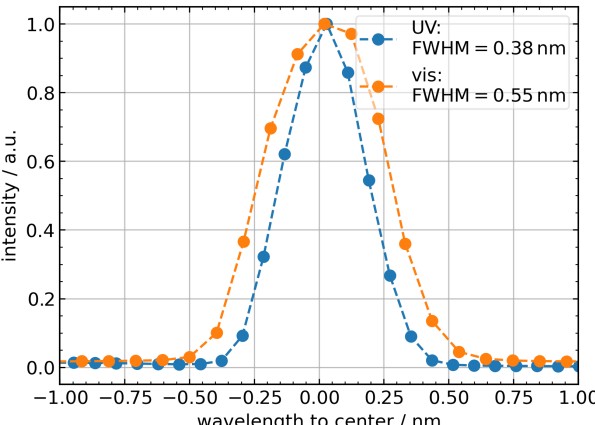

**Figure 3.** Spectral response function of the UV and vis spectrometers monitored through the mercury emission peak at 334 nm (blue) and the krypton peak at 432 nm (orange), respectively. The full-width at half maximum (FWHM) of the spectral response functions gives the resolution of the respective spectrometer. For both spectrometers the sampling is about 5 pixels per FWHM. The shown spectral response functions were recorded a few hours before the balloon launch from Timmins.

the balloon gondola for these two deployments, the available azimuth viewing range reduced to roughly 270° as the balloon and the gondola obscured the view of the sun in some directions.

Two telescopes, one optimised for the UV (2.54 cm (1 inch) diameter, 20 mm side length right-angle prism) and the other for the visible (1.27 cm (0.5 inch) diameter, 10 mm side length right-angle prism) spectral range, respectively, are placed into the parallel light beam between the azimuth mirror and the folding mirror, to detract sunlight for the spectral analysis by the spectrometer unit. The telescope design follows the design reported in Butz et al. (2017). By internal total reflection, the right-angle prism guides the light into the telescope tube with a coloured glass filter (UV: Hoya U340; vis: Hoya B460) for the respective desired wavelength ranges. Inside the telescope, a lens with a focal length of $f = 40$ mm (UV: Thorlabs LB4030-UV; vis: Thorlabs LB1378) focuses the light onto an aperture with a diameter of 800 μm to limit the field of view to 1.15° i.e. roughly corresponding to twice the diameter of the solar disk. Behind the aperture, the expanding light beam illuminates a diffuser plate at 45° angle and the light scattered off the diffuser plate is collected by a mono 400 μm glass fibre (numerical aperture of 0.22, length 3 m) which guides the light into the spectrometer.

## 3.2 Spectrometer unit

The spectrometer unit is constructed similarly to previous balloon-borne DOAS instruments, e.g. as documented by Ferlemann et al. (1998, 2000) and Weidner et al. (2005). For our setup, two commercial grating spectrometers (Ocean Insight QE-Pro), one for the UV (305 nm to 387 nm) and one for the vis (398 nm to 502 nm) spectral range, are used to record the solar absorption spectra. The UV spectrometer has a spectral resolution of 0.38 nm while the vis spectrometer has a spectral resolution of



0.55 nm. The spectral resolution of both spectrometers corresponds to $\sim 5$ spectral pixels. The spectral response functions of both spectrometers are measured using spectral emission lamps and are shown in Fig. 3.

The spectrometer unit is depicted in Fig. 2(a). The glass fibres coming from the solar tracker unit feed the sunlight into the spectrometers via a custom-made vacuum feedthrough to the spectrometer entrance slits of $100\,\mu m$ width each. A vacuum chamber houses both spectrometers to avoid wavelength shifts through pressure changes and any condensation on the actively thermo-electrically cooled CCD detectors (operated at $-10\,°C$). The temperature of the spectrometers is stabilised by a mixture of ice and water (volume of around $8\,l$) within a vessel surrounding the vacuum chamber. A small water pump continuously mixes the ice-water bath to prevent temperature gradients from forming. About $3\,cm$ of open-cell foam insulate the vessel from the outside. The spectrometer unit measures $0.45\,m \times 0.4\,m \times 0.4\,m$, weights around $25\,kg$ (including the water), and consumes $30\,W$.

The spectral acquisition software was developed in our lab. During stratospheric balloon flights, it can be operated via a remote connection to the embedded computer. Single spectra are stored locally on the embedded computer. Integration times between $10\,ms$ to $60000\,ms$ are adjusted manually during sunset and sunrise when illumination conditions change quickly.

### 3.3 Instrument performance

Overall, the instrument performed well during both deployments and a continuous set of spectra was acquired during the sunset for both flights.

The thermal budget of the instrument's electronic components was well balanced throughout both deployments except for the ascent of the flight from Kiruna. During this first deployment, the onboard computer integrated into the spectrometer unit shut down during balloon ascent due to low temperatures ($\sim 215\,K$) near the tropopause. However, radiative heating in the middle stratosphere brought the computer online again and high computational usage during night resulted in continuous measurements during the entire measurement period at balloon float altitude. In general, the computers were more threatened to get too cold than too warm. Therefore, two heaters were built into the onboard computers for the second deployment at Timmins, supplying up to $3\,W$ of heat when the computers' temperatures were below $0\,°C$. Additionally, the computational power was increased on purpose by dummy scripts during the critical phases to generate heat.

The performance of the spectrometers is mainly dependent on their thermal stability, which is governed by the stability of the ice water bath. Stable temperatures were measured with a PT1000 sensor within the ice-water bath throughout both flights with deviations lower than $\pm 0.25\,°C$. Nevertheless, a wavy pattern of the signal recorded by individual detector pixels is observed over time in the spectra acquired during both deployments, even during periods where all relevant parameters (integration time, relative azimuth angle of the sun to solar tracker, number of scans, etc.) were kept constant (as can be seen in Fig. 5). This artefact is most likely caused by some condensate on the detector, resulting in etalon structures. The optical density of these intensity changes is on the order of $10^{-3}$, similar to the BrO absorption structures. A promising correction method is described in Sect. 4.2. Unfortunately, this method can only be applied to the spectra recorded during the second deployment from Timmins because the relative azimuth angle of the tracker to the sun was kept constant during the entire sunset, resulting in a smooth intensity time series. However, this relative azimuth angle was changed several times during the first flight from




Kiruna, resulting in small changes in the recorded intensity, making a correction via polynomial fits unfeasible. Hence, all
results presented in the following sections are based on the spectral retrieval of the UV data from the Timmins deployment
only.

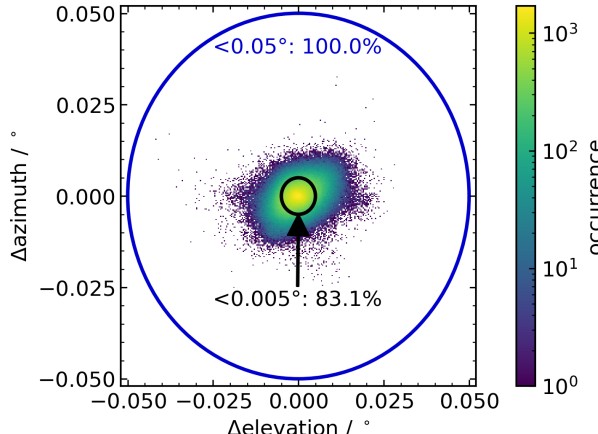

**Figure 4.** Tracking precision during sunset ($53° \leq$ SZA $\leq 95°$) of the flight from Timmins in 2022. The blue circle shows the target precision
of $0.05°$, whereas the black circle shows a deviation by $0.005°$ from the center pixel. All data points of the tracking during this timespan lie
within the target range and $83.1\%$ even deviate less than $0.005°$.

During the flight, the tracker needs to not only follow the sun's position in the sky but also compensate for any gondola
movement. During ascent, these movements of the gondola are the greatest with more than $\pm4°\text{s}^{-1}$. Once the gondola is
azimuth stabilised, it reduces drastically to angular velocities of less than $\pm1°\text{s}^{-1}$. The tracker was designed to compensate
for angular velocities of up to $\pm2°\text{s}^{-1}$. Therefore, during both deployments the tracker followed the sun sufficiently once the
gondola was stabilised in the azimuth direction. During ascent, when the gondola was not stabilised in azimuth, tracking the sun
was not continuously possible, and the balloon frequently obscured the tracker's view of the sun. The tracking precision during
the measurement time of the sunset from balloon float altitude ($53° \leq$ SZA $\leq 95°$) for the flight from Timmins is depicted by
the 2D histogram shown in Fig. 4. The deviation from the target position on the screen viewed by the fine-tracking camera
is given in degrees of the azimuth and elevation angle. All data points lie within the target range of $0.05°$, and $83.1\%$ of the
data points show deviations of less than $0.005°$. A similar precision with a few more outliers was achieved during the first
deployment from Kiruna.

## 4  Trace gas analysis

The recorded solar absorption spectra are first preprocessed to correct for instrument related effects which include an etalon
structure that needs particular attention (Sect. 4.1). Then, we use the DOAS method (Platt et al., 2008) to infer, for each
spectrum, the differential slant column densities (dSCDs) of the fitted trace gases. Here, we focus on the retrieval of BrO



dSCDs from the UV spectral range (Sect. 4.2) since the primary goal of the mission is to monitor bromine abundances in the mid-stratosphere. The instrument setup also supports the detection of IO, OClO, HONO, $O_3$ and $NO_2$ retrievals which are, however, not detailed here. To convert dSCDs into a mean BrO volume mixing ratio (VMR) above balloon altitude ([BrO]),
we employ Langley's method entailing the simulation of the light path from the sun to the balloon-borne instrument (Sect. 4.3). Finally, we estimate [$Br_y$] by simulating the bromine partitioning in the mid-stratosphere using a photochemical box model (Sect. 4.4).

## 4.1 Preprocessing

Prior to the balloon deployments, the nonlinearity of the spectrometers' detectors was characterised in lab measurements. The detector's response is most linear for pixel saturations between 30 % and 60 %, hence the integration time during the deployments was manually adjusted such that the maximum saturation is within this range. The detector's dark current and offset voltage as well as mercury and krypton emission lamp, and halogen lamp spectra were recorded a few hours before and a few hours to a day after each balloon deployment depending on the recovery time of the gondola.

As a first step of data preprocessing, the single spectra recorded during the flight are corrected for the detector's nonlinearity and 200 spectra are coadded to increase the signal-to-noise ratio. Further, the spectrometer's spectral response function used for convolving absorption cross sections is calculated from the emission lamp spectra. Here, the spectral emission line closest to the retrieval window is used for the convolution. Further, a wavelength pre-calibration for the DOAS evaluation is calculated from the spectra of the spectral emission lamps. The software used for the DOAS retrieval, QDOAS (Danckaert et al., 2012), corrects all spectra for the offset and dark current and performs a more advanced wavelength calibration using the solar Fraunhofer lines.

First DOAS retrievals showed the presence of disturbing residual structures in the retrieved spectra, resulting in a highly varying retrieval quality, i.e. the root-mean-square (RMS) of the spectral residuals of the DOAS fits showed an oscillating pattern over time. Further investigation indicated oscillations in the signal recorded by each detector pixel over time, but with different phases for individual pixels as depicted in Fig. 5. These oscillating signals are most likely caused by condensate on the detector resulting in spectral etalon structures. Given the oscillating pattern of the RMS, the effects of the etalon reproduce with time, i.e. same spectral structures are repeating themselves in each full oscillation. Therefore, we use a few oscillations to characterize the artefacts and to correct for them within the DOAS retrieval.

The overall correction strategy is to select a period where the SCDs of the target gases are safely below the detection limit and which is long enough to cover a few cycles of the oscillating pattern. From spectra recorded during that period, we extract the artefact signal in terms of an etalon optical density and characterize its spectral pattern through a principal component analysis (PCA). The inferred principal components are then used to correct all spectra including those used for scientific analysis within the DOAS retrieval. As the period for calibrating the principal components, we take the one corresponding to the SZA range between 55° and 75° when the balloon was at float altitude, the sun was high in the sky and anticipating that the DOAS retrievals use the spectrum recorded at SZA = 74.1° as their reference spectrum. Thus, the target gases show negligible spectral signals for the chosen period. Yet, the period covers five cycles of the oscillations.





In order to infer the etalon optical density $\tau_{\text{etalon}}$ from the spectra recorded during the calibration period, we first calculate the measured optical density $\tau_{\text{meas}}$ with respect to the reference spectrum (at SZA = 74.1°). The atmospheric contribution $\tau_{atmo}$ to $\tau_{\text{meas}}$ is mainly due to Rayleigh scattering which we approximate by a low order polynomial function P of either the slant airmass $\text{SCD}_{\text{air}}$ (with respect to the reference spectrum) or wavelength (see the two approaches below). The etalon optical density $\tau_{\text{etalon}}$ can thus be calculated from the measured optical density $\tau_{\text{meas}}$ via

$$\tau_{\text{etalon}}(\lambda, \text{SCD}_{\text{air}}) = \tau_{\text{meas}}(\lambda, \text{SCD}_{\text{air}}) - \tau_{\text{atmo}}(\lambda, \text{SCD}_{\text{air}}) = \tau_{\text{meas}}(\lambda, \text{SCD}_{\text{air}}) - \text{P}(\lambda, \text{SCD}_{\text{air}}). \tag{1}$$

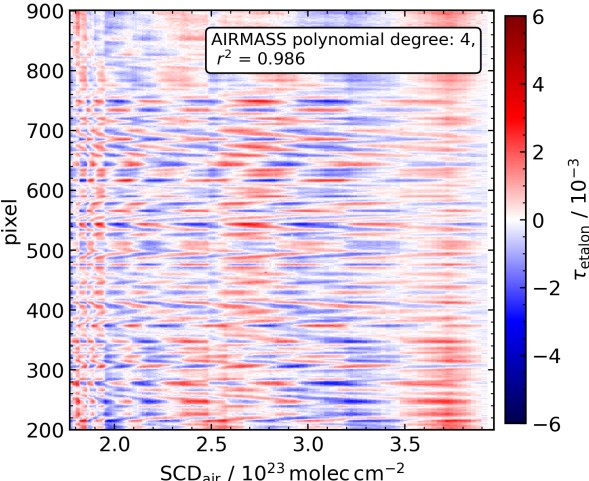

**Figure 5.** Optical density fluctuations due to etaloning on the CCD detector $\tau_{\text{etalon}}$ shown for detector pixels 200 to 900 as a function of slant airmass $\text{SCD}_{\text{air}}$. The spectral pattern was obtained using the approach AIRMASS with a $4^{\text{th}}$ degree polynomial as described in the text. The chosen reference spectrum at SZA = 74.1° has a $\text{SCD}_{\text{air}}$ of $3.9e23\,\text{molec}\,\text{cm}^{-2}$. The mean $r^2$ value for the polynomial fits over all pixels is 0.986.

Then, there are two approaches on how to implement Eq. 1. The first approach (further called AIRMASS) finds the atmospheric optical density $\tau_{atmo}$ by fitting the polynomial coefficients of $\text{P}(\text{SCD}_{\text{air}})$ (between third and fifth order) to the measured optical density $\tau_{\text{meas}}$ for each detector pixel as a function of slant airmass $\text{SCD}_{\text{air}}$. The etalon optical density $\tau_{\text{etalon}}$ resulting from this procedure is shown in Fig. 5. The second approach (further called SPEC) finds the atmospheric optical density $\tau_{atmo}$ by fitting a third order polynomial $\text{P}(\lambda)$ to the measured optical density $\tau_{\text{meas}}$ for each spectrum as a function of wavelength $\lambda$. Using the SPEC approach, the results for $\tau_{\text{etalon}}$ are similar to the oscillations shown in Fig. 5 obtained using the AIRMASS approach.

No matter what approach is used, a PCA is then performed on $\tau_{\text{etalon}}(\lambda, \text{SCD}_{\text{air}})$ to retrieve the dominating spectral features, i.e. the principle components (PCs) as a function of wavelength $\lambda$ over the calibration period (with time parameterized through $\text{SCD}_{\text{air}}$). The PCs are sorted by the variance they explain along the $\text{SCD}_{\text{air}}$ dimension, and the first 5 to 7 PCs are finally included in the DOAS analysis by including them in the DOAS fits as pseudo-absorbers. As various settings (approach





**Table 1.** Absorption cross sections used for the BrO DOAS fit.

| Gas | Temperature | SCD for Solar $I_0$ | Citation |
|-----|-------------|---------------------|----------|
| $O_3$ | 203 K and 246 K | $5e19$ molec cm$^{-2}$ | Voigt et al. (2001) |
| $NO_2$ | 246 K | $5e16$ molec cm$^{-2}$ | Voigt et al. (2002) |
| $O_4$ | 253 K | - | Thalman and Volkamer (2013) |
| BrO | 223 K or 298 K | - | Wahner et al. (1988) |

AIRMASS and SPEC, order of polynomials, number of PCs) used for the etalon correction yield similar results, a sensitivity study is conducted to estimate the uncertainty propagated by the respective choices into the final results.

## 4.2 Spectral retrieval of BrO

We infer differential slant column densities of BrO with respect to a reference spectrum by applying the established DOAS technique (Platt, 1994; Platt et al., 2008) using the QDOAS software (Danckaert et al., 2012). Spectrally interfering species are $O_3$, $NO_2$, and $O_4$. The reference is chosen from the collection of spectra recorded at balloon float altitude when the slant column absorption is minimal (here SZA = 74.1°).

Following the recommendations of Aliwell et al. (2002) and previous balloon-borne BrO studies (Harder et al., 2000; Dorf et al., 2006b; Kreycy et al., 2013; Rotermund et al., 2021), the retrieval window extends over the range 346 nm to 360 nm covering two BrO absorption bands. All absorption cross sections used for the DOAS fit are listed in Table 1. An additive offset polynomial (order 1) is included to account for instrumental stray light and a 2$^{nd}$ degree polynomial is included to account for any broadband extinction processes. To make the results of this study comparable with other balloon-borne stratospheric BrO

measurements (Harder et al., 2000; Dorf et al., 2006b; Kreycy et al., 2013), the DOAS retrieval settings are consistent with these studies and the BrO absorption cross section by Wahner et al. (1988) is used for the retrieval with a fixed wavelength shift of 0.28 nm (Wilmouth et al., 1999). All absorption cross sections are convolved to the instrument's spectral resolution using the measured spectral response function shown in Fig. 3. As the absorption cross section of $O_3$ is temperature dependent, absorption cross sections for two different temperatures are included in the fit, with the absorption cross section at 203 K being

orthogonalised to the one at 246 K. The absorption cross sections of the strong absorbers $O_3$ and $NO_2$ are also corrected for the solar $I_0$ effect using the SCDs listed in Table 1 (Aliwell et al., 2002; Platt et al., 2008). Additionally, five to seven principle components are included to represent the etalon structures determined during the preprocessing in the DOAS retrieval as pseudo-absorbers. Figure 6 illustrates the spectral fitting quality.

Since the absorption cross section of BrO is approximately linearly dependent on the gas temperature, the BrO dSCD of

each spectrum is calculated from an interpolation between the dSCDs retrieved with the two BrO absorption cross sections recorded at $T = 223$ K (cold) and $T = 298 K$ (warm) (Wahner et al., 1988) to the effective absorption temperature.

The effective absorption temperature of BrO as function of SZA and observation altitude $T_{\text{eff}}(SZA)$ can be inferred from the photochemical simulations (Sect. 4.4) and the simulated light path yielding the air mass factor (AMF) and the vertical

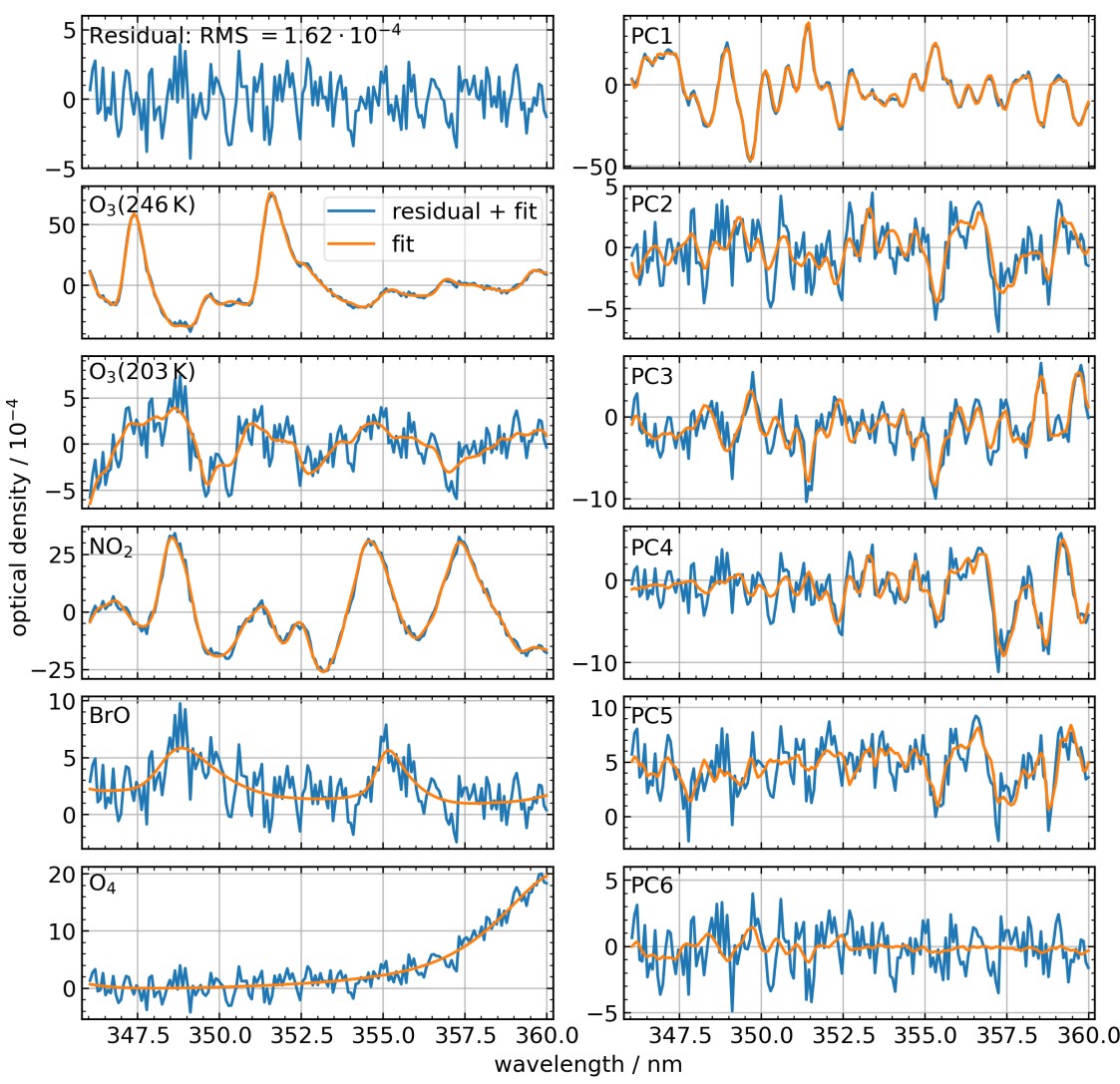

**Figure 6.** DOAS fit results for the spectrum recorded at SZA = 89.9°. The spectral retrieval uses two $O_3$ absorption cross sections at different temperatures, a $NO_2$ absorption cross section, the $BrO$ absorption cross section at 298 K, an $O_4$ absorption cross section and the first six PCs representing the etalon structures calculated via the AIRMASS approach described in the text. In addition to the fit components shown here, the DOAS retrieval adopts a first order offset polynomial and a $2^{nd}$ order polynomial to account for broadband extinction processes.



density VD (Sect. 4.3). It is calculated via

$$T_{\text{eff}}(\text{SZA}) = \frac{\sum_{i=1}^{L} \text{AMF}_i(\text{SZA}) \cdot T_i \cdot [\text{BrO}]_i \cdot \text{VD}_i}{\sum_{i=1}^{L} \text{AMF}_i(\text{SZA}) \cdot [\text{BrO}]_i \cdot \text{VD}_i} \tag{2}$$

with the temperature $T_i$ and BrO VMR [BrO] in each layer $i$.

### 4.3 Langley's method

Under the assumption of a constant [BrO] within the altitude range from the balloon to a few kilometers above, the BrO dSCDs
(dSCD$_{\text{BrO}}$) should scale linearly with the slant column density of air (SCD$_{\text{air}}$) (Langley, 1904; Bösch et al., 2001; Dorf et al.,
2006b),

$$\text{dSCD}_{\text{BrO}} = [\text{BrO}] \cdot \text{SCD}_{\text{air}} - \text{SCD}_{\text{BrO},0} \tag{3}$$

where SCD$_{\text{BrO},0}$ is the slant column of BrO in the reference spectrum. SCD$_{\text{air}}$ is calculated by our ray-tracing program
DAMF which has been validated in the studies of Harder et al. (2000); Bösch et al. (2001); Dorf et al. (2006b); Butz et al.
(2009); Weidner et al. (2005). DAMF calculates the box air mass factor AMF$_i$ for each layer $i$, i.e. the factor by which the
air mass along the slant light path is larger compared to the vertical column density of air in the atmospheric layer of height
$\Delta h_i \geq 100\,\text{m}$. The AMF calculation is based on a spherical atmosphere and takes into account refraction due the changing
pressure in the atmosphere. The used atmospheric pressure and temperature profile is based on ERA5 data (Hersbach et al.,
2020, 2023) and above the highest ERA5 altitude of $1\,\text{hPa}$ (corresponding to about $48.5\,\text{km}$) the US standard atmosphere
(scaled to the ERA5 profile) is used. The resulting SCD$_{\text{air}}$ is then obtained for each spectrum from

$$\text{SCD}_{\text{air}} = \sum_{i=1}^{L} \text{AMF}_i \cdot \text{VD}_i \cdot \Delta h_i \tag{4}$$

using the air vertical column density VD$_i$ of layer $i$.

Using Langley's method according to Eq. 3 , i.e. performing a linear regression to the BrO dSCDs vs. SCD$_{\text{air}}$, provides the
[BrO] above balloon float altitude and SCD$_{\text{BrO},0}$.

In former studies a representative partitioning ratio was used to scale the Langley-derived [BrO] to the Br$_{\text{y}}$ VMR [Br$_{\text{y}}$].
However, this assumption is only approximately valid. Therefore, in the present study, Eq. 4 is modified to

$$\text{SCD}_{\text{air, weighted}} = \sum_{i=1}^{L} \text{AMF}_i \cdot \left( \frac{[\text{BrO}]}{[\text{Br}_{\text{y}}]} \right)_i \cdot \text{VD}_i \cdot \Delta h_i. \tag{5}$$

to account for the $\frac{[\text{BrO}]}{[\text{Br}_{\text{y}}]}$ ratio varying with height and with the local SZA. The respective $\frac{[\text{BrO}]}{[\text{Br}_{\text{y}}]}$ ratio as a function of altitude and
SZA is obtained from photochemical modelling (Sect. 4.4). Reconsidering Langley's method with the partitioning-weighted
SCD$_{\text{air, weighted}}$ on the abscissa, [Br$_{\text{y}}$] can be inferred directly from the slope of a linear regression to the BrO dSCDs.

### 4.4 Photochemical modelling

At daytime, BrO is the predominat bromine species in the middle stratosphere. Due to the well-mixed conditions in the middle
stratosphere, the major bromine species rapidly approach steady state under sunlight. The main reactions governing the bromine





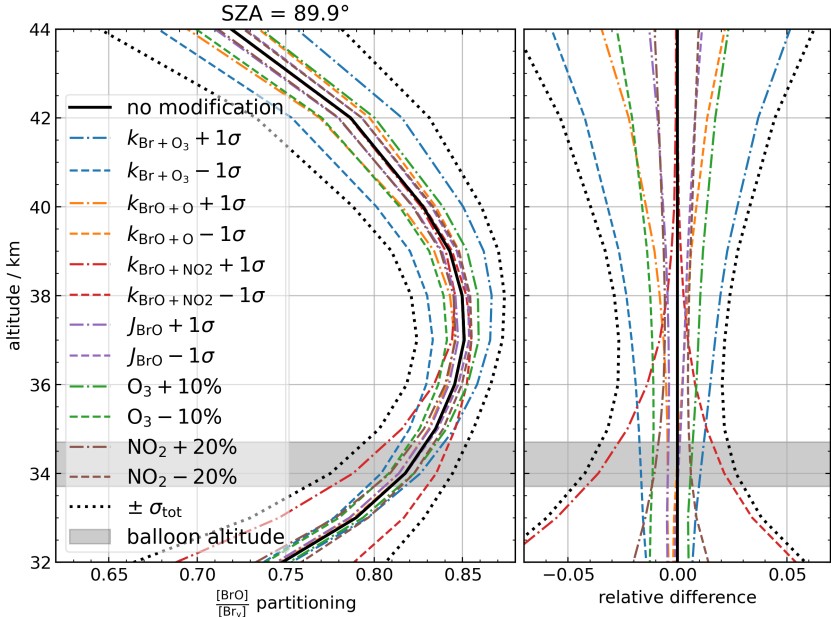

**Figure 7.** The $\frac{[\mathrm{BrO}]}{[\mathrm{Br}_y]}$ partitioning at $89.9°$ SZA at $30\,\mathrm{km}$ to $45\,\mathrm{km}$ altitude (left panel). In addition to the model base run (black), we conducted several simulations under perturbed model parameters (colored) as indicated by the legend and as described in the main text. The right panel shows the relative difference of the perturbed simulations to the base run. The total uncertainty (black dotted) is calculated by Gaussian summation of the individual differences to the base run. The light grey altitude range marks the balloon gondola's altitude during the observations used for the regression in the Langley plot ($33.7\,\mathrm{km}$ to $34.7\,\mathrm{km}$).

partitioning in the middle stratosphere are the inter-conversion reactions between the three most reactive bromine species $\mathrm{BrO}$, $\mathrm{BrONO}_2$, and $\mathrm{Br}$ via reactions with odd oxygen species

$$\mathrm{Br} + \mathrm{O}_3 \rightarrow \mathrm{BrO} + \mathrm{O}_2 \tag{6}$$

and

$$\mathrm{BrO} + \mathrm{O} \rightarrow \mathrm{Br} + \mathrm{O}_2, \tag{7}$$

the photolysis of bromine oxide,

$$\mathrm{BrO} + \mathrm{h}\nu(\lambda \leq 515\,\mathrm{nm}) \rightarrow \mathrm{Br} + \mathrm{O}, \tag{8}$$

and the formation of the predominat nighttime species bromine nitrate

$$\mathrm{BrO} + \mathrm{NO}_2 + \mathrm{M} \rightarrow \mathrm{BrONO}_2 + \mathrm{M}. \tag{9}$$

Here, $\mathrm{M}$ represents a third but inert collision partner needed for momentum conservation.





We use a 1D photochemical stacked box (column) model to calculate the time evolution of stratospheric inorganic bromine species. The chemical mechanism is based on the off-line chemical transport model TOMCAT (Chipperfield, 1999, 2006)

similar to the setup in Höpfner et al. (2021) and Sinnhuber et al. (2005) with all bi-molecular and ter-molecular rate constants being updated to JPL2022 (Burkholder et al., 2019). The model simulates the chemical evolution of stratospheric species over the course of five days, with the flight day being the last day. The model calculates actinic fluxes in a pseudo-spherical geometry with the direct solar beam calculated taking into account the sphericity of the atmosphere and multiple scattering calculated in plane-parallel approximation, using a scheme based on Lary and Pyle (1991), which in turn is based on Meier et al. (1982)

and Nicolet et al. (1982). The actinic flux calculations of the model have been validated by Bösch et al. (2001) for similar balloon borne observations. VMR profiles are calculated at the balloon-launch site's latitude and longitude at 47 vertical levels between $10\,\mathrm{km}$ to $132\,\mathrm{km}$. Initial temperature and pressure profiles are taken from ERA5 data. The $O_3$ profile is initialised by a microwave limb sounder (MLS) ozone profile recorded in close temporal and spatial vicinity to the balloon flight (Schwartz et al., 2020). Other gas profiles are initialised by EMAC simulations presented in Höpfner et al. (2021).

The simulations are used to infer the [BrO] to total bromine $[Br_y]$ partitioning as a function of altitude and SZA (shown in black in Fig. 7 for SZA = 89.9°). For $80° \leq SZA \leq 90°$ this partitioning is between $0.8$ and $0.9$ for altitudes between $33\,\mathrm{km}$ and $41\,\mathrm{km}$. The uncertainty of the partitioning profile is estimated by varying model parameters important for stratospheric bromine chemistry. These parameters include the rate constants $k$ of reactions (6), (7), and (9) each varied by $\pm 1\sigma$ at 250K according to Burkholder et al. (2019). Additionally, the photolysis frequency of BrO $J_{BrO}$, ( reaction 8) is varied by $\pm 8\,\%$

according to the uncertainty of the absorption cross section (Wilmouth et al., 1999) and the $O_3$ VMR is varied by $\pm 10\,\%$ while the initial $NO_2$ VMR (together with the NO VMR) is varied by $\pm 20\,\%$. The resulting profiles of the $\frac{[BrO]}{[Br_y]}$ partitioning are shown as dashed lines in Fig. 7. The combined uncertainty of all these runs gives the estimated modelling error of $\frac{[BrO]}{[Br_y]}$. In the relevant altitude range ($33.7\,\mathrm{km}$ to $40\,\mathrm{km}$), this error amounts to $5\,\%$ which agrees with a similar study in Dorf et al. (2006b). The largest contribution comes from reaction (9).

## 5 Stratospheric bromine abundance

Given the analytical tools outlined in Sect. 4, we quantify the stratospheric bromine abundance in terms of the mid-stratospheric [BrO] (Sect. 5.1), the compatible VMR of total gaseous inorganic bromine $[Br_y]$ (Sect. 5.2), and the implications for the bromine trend in the stratosphere (Sect. 5.3).

### 5.1 BrO abundance in the mid-stratosphere

The BrO dSCDs are retrieved from the measured absorption spectra as described in Sect. 4.2 including an interpolation to the effective absorption temperature according to Eq. 2. Thereby, the retrieval is run repeatedly with a variable set of preprocessing parameters (see Table A1) to estimate the uncertainty introduced by the etalon structures and their correction. For illustration purposes, we report on a particular setting (approach AIRMASS, 4th degree polynomial, 6 PCs) as being representative for the quality of our [BrO] estimate and we consider the spread of sensitivity runs within the error analysis.



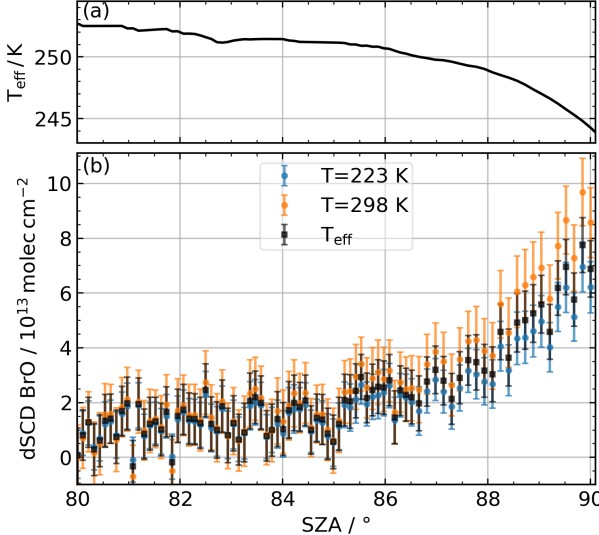

**Figure 8.** Panel (a) shows the BrO effective absorption temperature as a function of the SZA. Panel (b) shows the retrieved dSCDs of BrO for the two BrO absorption cross section temperatures 223 K (blue) and 298 K (orange) reported by Wahner et al. (1988) and the dSCDs of BrO interpolated to the effective absorption temperature (black).

Figure 8 shows the estimated BrO dSCDs for the measurement period with SZA ranging between $80°$ and $89.9°$. This period is outside the range used for calibrating the etalon correction, and the balloon is at float altitude ($33.7\,\mathrm{km}$ to $34.7\,\mathrm{km}$), the sun is approaching the horizon, implying a long light path through the overhead atmosphere, and the $\frac{[\mathrm{BrO}]}{[\mathrm{Br_y}]}$ partitioning does not change vastly. The latter allows for detecting the minor abundant BrO, which is clearly visible in the increase of the detected BrO dSCDs with SZA. Figure 8 also illustrates the effect of interpolating the BrO dSCDs from the retrieval nodes at 223 K
and 298 K to the effective absorption temperature. The error bars on the dSCDs represent the one sigma uncertainty of the DOAS retrieval.

    The error weighted Langley linear regression (Fig. 9) based on the BrO dSCDs (interpolated to the effective temperature) yields $[\mathrm{BrO}] = (14.1 \pm 1.0)\,\mathrm{ppt}$, which is representative for the mean BrO VMR above balloon altitude where the $[\mathrm{BrO}]$ uncertainty represents the statistical uncertainty of the linear regression. For the Langley regression, the SZA range between
$86.0°$ and $89.9°$ (corresponding to $1.6e24\,\mathrm{molec\,cm^{-2}} \leq \mathrm{SCD_{air}} \leq 5.3e24\,\mathrm{molec\,cm^{-2}}$) is used since for SZA$< 86.0°$ the BrO dSCDs are below the detection limit commonly taken as twice the dSCD retrieval uncertainty.

    To estimate possible errors introduced by the etalon correction during preprocessing, we perform a sensitivity study with different choices of preprocessing parameters as listed in Table A1. The BrO dSCDs and the Langley regressions are calculated for each parameter configuration (see Fig. A1 and B1). For the ensemble of sensitivity runs, the inferred mid-stratospheric
$[\mathrm{BrO}]$ ranges between $(12.7 \pm 1.0)\,\mathrm{ppt}$ and $(16.4 \pm 1.0)\,\mathrm{ppt}$ while the quality of the spectral retrieval is comparable among all runs. The inferred $[\mathrm{BrO}]$ is similar across different numbers of PCs included in the DOAS fit, however, $[\mathrm{BrO}]$ is significantly different for different polynomial degrees. None of the preprocessing settings seems to produce significantly different results





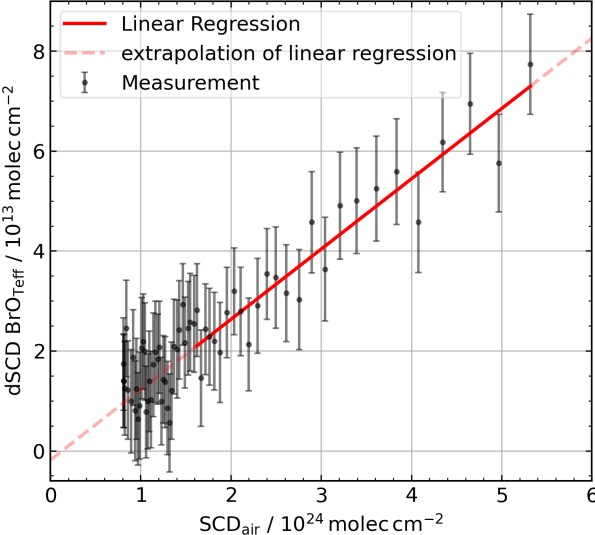

**Figure 9.** Langley plot for BrO. The dSCD BrO error bars indicate the $1\sigma$ DOAS retrieval uncertainty. The linear regression (red solid line) yields a slope of $[\mathrm{BrO}] = (14.1 \pm 1.0)$ ppt and a y-axis intercept of $\mathrm{SCD}_{\mathrm{BrO},0} = (1.8 \pm 3.0)10^{12}$ molec cm$^{-2}$. The faint dashed red line indicates the extrapolation of the linear regression outside the fit range of $86° \leq \mathrm{SZA} \leq 89.9°$.

concerning the RMS of the DOAS residuals or dSCD dependencies on SZA. Thus, we integrate all runs into our final result by calculating the mean and the standard deviation among the ensemble. The mean mid-stratospheric [BrO] over the ensemble

(Fig. B1) is 14.4 ppt with a standard deviation of 1.3 ppt on top of a mean Langely regression error of 1.0 ppt. Note that the statistical treatment of the preprocessing error estimated by the standard deviation over the ensemble is debatable since it is actually of systematic nature. In absence of better insight, we nevertheless use Gaussian error propagation to combine the preprocessing and Langley-fit error contributions, which yields a total error of 1.6 ppt (11 %). Additionally, the systematic error of the BrO absorption cross section (about 8%) needs to be added when comparing with studies using different BrO

absorption cross sections.

### 5.2  Br$_\mathrm{y}$ in the mid-stratosphere

[Br$_\mathrm{y}$] is inferred from the retrieved BrO dSCDs using the air mass factor matrix and the model-based $\frac{[\mathrm{BrO}]}{[\mathrm{Br_y}]}$ ratio as described in Sect. 4.3. Figure 10 shows the respective Langley regression for the same illustrative sensitivity run as in Sect. 5.1. The linear regression of BrO dSCDs against slant airmass yields [Br$_\mathrm{y}$] = 17.0 ppt with a fit error of 1.2 ppt.

The Langley regressions of all sensitivity runs are depicted in Fig. B2. Among the ensemble, [Br$_\mathrm{y}$] ranges between 15.4 ppt and 19.9 ppt. The mean [Br$_\mathrm{y}$] among all sensitivity runs is 17.5 ppt with a standard deviation of 1.6 ppt (9 %). The mean Langley-fit uncertainty among all runs is 1.2 ppt (7 %). Additionally, we need to consider the uncertainty of the $\frac{[\mathrm{BrO}]}{[\mathrm{Br_y}]}$ ratio inferred from photochemical simulations which amounts to 5 % (as described in Sect. 4.4). Again resorting to Gaussian error propagation for combining all three error sources, the total uncertainty of the inferred [Br$_\mathrm{y}$] is 2.2 ppt (12 %). Further, we need



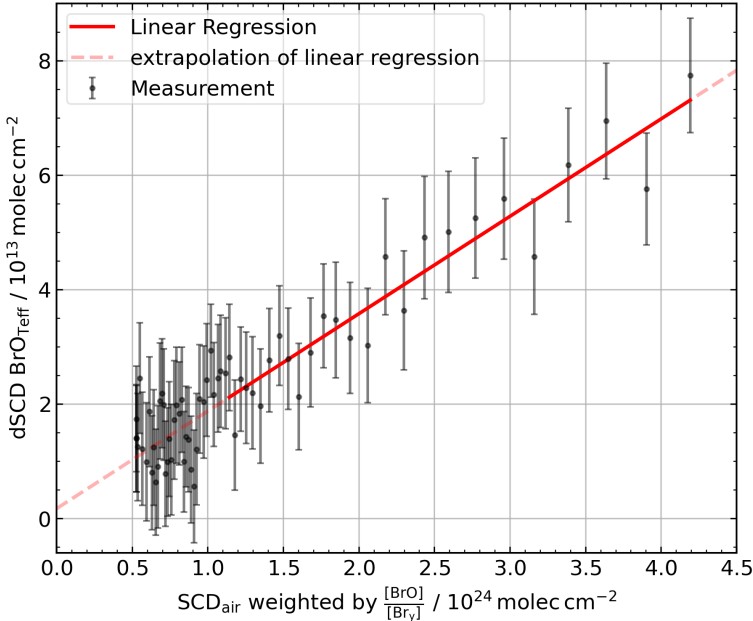

**Figure 10.** Langley plot for $Br_y$. A linear regression (red solid line) is fitted to the BrO dSCDs of BrO against the $SCD_{air}$ weighted by $\frac{[BrO]}{[Br_y]}$ ratio as described in the text. The slope of the fit yields $[Br_y]$ of $(17.0 \pm 1.2)$ ppt. Again, the BrO dSCDs of spectra with SZA < 86° are not included in the fit (red dashed line) since they are lower than the detection limit of our instrument (twice the DOAS fit error).

to add the systematic uncertainty of the BrO absorption cross section of 8 % when our $[Br_y]$ is compared with results based on a different absorption cross sections or a totally different method.

## 5.3 Inferred $Br_y$ in context of bromine trend

Our measurements result in $[Br_y] = (17.5 \pm 2.2)$ ppt with a combined uncertainty of statistical (1.5 ppt) and systematic (1.6 ppt) uncertainty plus an additional 8 % contribution from the absorption cross section needed when compared to re-
sults using a different absorption cross section. To compare this result with the published trend in total stratospheric bromine (Laube et al., 2022), we need to determine the year of stratospheric entry of the observed airmasses. To this end, we estimate the age of air from $N_2O$ VMRs either measured in close temporal and spatial vicinity by the MLS instrument (Waters et al., 2006) or from the co-deployed GLORIA instrument (see Fig. C1). Using the $N_2O$ to age of air relationship of Engel et al. (2002), and updating the relationship with present $N_2O$ VMRs at stratospheric entry level (Lan et al., 2022), we infer a mean
age of the probed air masses of $(5.5 \pm 1.0)$ yrs. This results in a stratospheric entry date in early $2017 \pm 1$ yr.

Given this date and the size of our errors, our $[Br_y]$ estimate fits reasonably well into the trend assessments provided in Figure 1-14(a) of the recent scientific assessment of ozone depletion (Laube et al., 2022). Moreover, Rotermund et al. (2021) inferred a total stratospheric bromine mixing ratio $[Br^{tot}]$ from total inorganic bromine $Br_y$ and organic bromine species of $[Br^{tot}] = (19.2 \pm 1.2)$ ppt for air younger than about 1 yr in the lower stratosphere measured from aboard the HALO aircraft





during the WISE campaign in fall 2017. Since there should be no sizeable organic bromine abundance present in the mid-stratosphere, our inferred $[\mathrm{Br_y}]$ can be compared directly to their $[\mathrm{Br^{tot}}]$. The estimate by Rotermund et al. (2021) is greater by $1.7\,\mathrm{ppt}$ than ours but agrees within our comparatively large error bar.

However, in the future, it should be feasible for us to infer $[\mathrm{Br_y}]$ with higher accuracy than presented here, which would add more information to the trend in stratospheric bromine when continued over a longer period. The predominant contribution

to the uncertainty of our $\mathrm{Br_y}$ estimate are the error arising from the etalon correction, the error of the $\mathrm{BrO}$ absorption cross section, and the noise error of the retrieved $\mathrm{BrO}$ dSCDs.

We plan to replace the CCD detectors that were contaminated in the past, which should eliminate the detected etalon structures in the spectra and thus, the respective error contribution. Furthermore, the noise in the dSCDs can be reduced by enhancing the light throughput. Our instrument setup could accommodate a factor 2 larger light input into the spectrometers by choosing

different glass fibres, which should reduce the noise by a factor of $\sqrt{2}$ for the same temporal coadding. Both measures could reduce the error of the inferred $[\mathrm{Br_y}]$ of our measurements. Our assessments would also benefit from a better constrained $\mathrm{BrO}$ absorption cross section also with respect to its temperature dependence, which is used to retrieve $\mathrm{BrO}$ using the DOAS method as well as in the photochemical simulation of the $\frac{[\mathrm{BrO}]}{[\mathrm{Br_y}]}$ ratio. Further, better constrained reaction rate constants for the dominant daytime bromine reactions (reactions (6), (7), and (9)) at stratospheric temperatures would decrease the uncertainty

on the estimated $\frac{[\mathrm{BrO}]}{[\mathrm{Br_y}]}$ ratio. More information on the photochemistry of stratospheric bromine could be obtained from simultaneous balloon-borne measurements of $\mathrm{BrO}$ (e.g. our balloon-borne DOAS instrument) and $\mathrm{BrONO_2}$ (MIPAS or GLORIA instrument, Wetzel et al. (2017); Höpfner et al. (2021)). In fact, for both deployments reported here, we intended joint measurements of $\mathrm{BrO}$ and $\mathrm{BrONO_2}$, but the data coverage (e.g. lacking ascent data, see e.g. Dorf et al. (2008)) and quality of our measurements during the previous deployments are still not as good as desired for such a study.

If operational issues currently preventing us from continuous observations during balloon ascent can be solved, we should be capable of extending our analyses from merely using balloon float data to inferring vertical trace gas profiles of $\mathrm{BrO}$, $\mathrm{O_3}$, $\mathrm{NO_2}$, and possibly $\mathrm{HONO}$ from the upper troposphere/lower stratosphere into the mid-stratosphere similarly to previous balloon-borne deployments (Dorf et al., 2006a; Butz et al., 2006; Kritten et al., 2010).

Further, the instrument potentially offers the opportunity to measure mid-stratospheric $\mathrm{IO}$ abundances using the spectra

recorded by the vis spectrometer. Solar occultation measurements from balloon-float altitudes of $35\,\mathrm{km}$ at $\mathrm{SZA} \approx 95°$ provide very long light paths through the stratosphere and thus the possibility to detect even very low $\mathrm{IO}$ abundances (Bösch et al., 2003; Butz et al., 2009).

## 6 Conclusion

We have developed a new balloon-borne solar occultation DOAS instrument designed for measurements of UV/vis absorbing

gases mainly relevant to ozone chemistry in the stratosphere such as $\mathrm{O_3}$, $\mathrm{NO_2}$, $\mathrm{BrO}$, and possibly $\mathrm{IO}$, $\mathrm{OClO}$, and $\mathrm{HONO}$. The instrument is of medium weight ($< 40\,\mathrm{kg}$) and has a low power consumption ($< 100\,\mathrm{W}$), making it suitable as a secondary instrument on azimuth-controlled balloon gondolas. Its modular design combines a stand-alone solar tracker with two





temperature and pressure stabilised optical spectrometers coupled through glass fibre bundles. The solar tracker is capable of compensating for the rotational motion of the gondola up to a rate of $\pm 2\,°\,\mathrm{s}^{-1}$ via alt-azimuth mounted mirrors controlled by
a PID control loop.

Two deployments of the instrument onboard the azimuth-stabilised gondola HEMERA offered by CNES (Centre National d'Etudes Spatial) were conducted in 2021 and 2022 from Kiruna, Sweden, and Timmins, Canada, respectively. The precision of the solar tracking system was excellent with virtually all measurements complying with the targeted precision of $0.05°$ with respect to the center of the solar disk ($\frac{1}{10}$ of sun's diameter) during balloon float in the middle stratosphere. However, due
to operational constraints, the gondola rotational motions during balloon ascent were too fast to allow for solar tracking. The optical spectrometers performed reasonably well apart from an oscillating spectral pattern that we attribute to etaloning on the CCD detectors. Nevertheless, for spectra acquired during the deployment in 2022, these etalon structures could be corrected for by a principle component analysis. After this correction, slant column densities of $O_3$, $NO_2$ and $BrO$ could be retrieved. Using Langley's method, the $BrO$ dSCDs are converted into mid-stratospheric (above $33.7\,\mathrm{km}$) $[BrO]$ VMRs which amounted
to $(14.4 \pm 1.6)\,\mathrm{ppt}$ with an additional systematic error due to the $BrO$ absorption cross section of $8\,\%$.

In order to estimate the total gaseous inorganic bromine load $Br_y$, we simulate the mid-stratospheric $\frac{[BrO]}{[Br_y]}$ ratio as function of altitude and SZA using a photochemical model based on TOMCAT. We find $[Br_y] = (17.5 \pm 2.2)\,\mathrm{ppt}$ plus $8\,\%$ contribution from the absorption cross section. The $[Br_y]$ uncertainty reflects the combined statistical ($1.5\,\mathrm{ppt}$) and systematic ($1.6\,\mathrm{ppt}$) uncertainty. Based on the age of air relationship driven by $N_2O$ profiles measured by the GLORIA instrument onboard the
same balloon gondola, the stratospheric entry date of the probed air masses is early $2017 \pm 1\,\mathrm{yr}$. Our inferred $[Br_y]$ fits reasonably well into the decadal stratospheric trend given by Laube et al. (2022) with a $[Bry]$ lower by $1.7\,\mathrm{ppt}$ than the total stratospheric bromine inferred by Rotermund et al. (2021) with a similar stratospheric entry date. Both results agree with each other considering the combined error bars.

Further improvements in our instrument will be directed toward reducing the errors by improving the light throughput and
eliminating the etaloning from the CCD detectors. Future deployments of the instrument and the respective data analysis will focus on exploiting the capabilities in terms of measuring the full suite of accessible chemical species such as IO to better constrain the stratospheric iodine load. Given its suitability as a secondary payload, we will aim at co-deploying the instrument with other stratospheric chemistry missions to contextualize the photochemical regime through various observational constraints.

*Data availability.* The data recorded by the balloon-borne DOAS instrument and the GLORIA $N_2O$ profile are available from the corresponding author upon request. The MLS data can be obtained via the Goddard Earth Sciences Data and Information Services Center archive, https://disc.gsfc.nasa.gov (last access $N_2O$ data: 20 March 2023, registration required, https://disc.gsfc.nasa.gov/datasets/ML3DBN2O_004/ summary, (Lambert et al., 2020); last access O3 data: 15 Febuary 2023, registration required, https://disc.gsfc.nasa.gov/datasets/ML2O3_ 005/summary, (Schwartz et al., 2020)). The ERA5 data can be obtained from the Copernicus Climate Data Store (last access: 11 January
2023, registration required, https://doi.org/10.24381/cds.bd0915c6, (Hersbach et al., 2023)).



## Appendix A: DOAS retrievals with different sets of preprocessing parameters

**Table A1.** Parameter configurations used for estimating the sensitivity of the BrO DOAS retrievals to the etalon correction in the preprocessing step.

| polynomial degree | Approach AIRMASS | Approach SPEC |
|:---:|:---:|:---:|
| 3 | 5 PCs, 6 PCs, 7 PCs | 6 PCs, 7 PCs |
| 4 | 5 PCs, 6 PCs, 7 PCs | - |
| 5 | 5 PCs, 6 PCs, 7 PCs | - |

The choice of the best parameters to use in the preprocessing to derive the pseudo-absorbers to be included in the DOAS retrieval is not straightforward. Thus, a sensitivity study is performed to investigate the impact of different sets of pseudo-absorbers. The following parameters are varied in this sensitivity study (Table A1): (a) approach AIRMASS or SPEC, (b) polynomial degree, and (c) number of included PCs in DOAS retrievals. In total 11 different DOAS retrievals are carried out for each of the two BrO absorption cross section temperatures.

The RMS and the dSCDs of $O_3$, $NO_2$, and BrO retrieved via the DOAS method for all 11 configurations are displayed in Fig. A1. Different colours and markers indicate the different sensitivity runs. In comparison to the retrieval with the cold BrO absorption cross section, the retrieval with the warm BrO absorption cross section yields similar RMS, $O_3$, and $NO_2$ dSCDs but slightly higher BrO dSCDs (shown in Fig. 8(b)). Figure A1 shows, that the retrieved dSCDs of ozone (Fig. A1(b)) and $NO_2$ (Fig. A1(c)) are not sensitive to the different sets of pseudo-absorbers included in the sensitivity runs, because the optical densities of these gases along the line of sight is larger than the spectral etalon structures. However, the RMS of the spectral residuals (Fig. A1(a)) and the BrO dSCDs (Fig. A1(d)) vary slightly for the different sensitivity runs. Nevertheless, the differences in the retrieved BrO dSCDs between the sensitivity runs are lower than the dSCD errors (indicated by error bars). Furthermore, the variance in the RMS over all spectra retrieved with one set of pseudo-absorbers is larger than the differences in the RMS between the different sensitivity runs of one spectrum. Additionally, the differences in the BrO dSCDs and RMS between two spectra are similar in magnitude and sign for all sensitivity runs, i.e. the lowest RMS of the residual is found for the same spectrum independent of the sensitivity run. Markers of the same colour mostly lie very close to each other while markers of different colours show larger discrepancies in the retrieved dSCDs. Thus, the effect of the chosen approach (AIRMASS or SPEC) and the polynomial order shows a higher impact on the retrieved dSCDs than the number of principle components included in the DOAS fits.

The BrO dSCDs as a function of the SZA shows slightly different slopes for the different sensitivity runs, i.e. the runs marked in red (AIRMASS with polynomial order 4) result in the lowest dSCDs at SZA between 89° and 90° whereas the runs marked in purple (AIRMASS with polynomial degree 5) result in the lowest dSCDS for nearly all spectra recorded at SZAs between 80° and 89°. In general, the dSCDs of the sensitivity runs marked in blue (approach SPEC) show the largest retrieved BrO dSCDs for all SZAs.





**Figure A1.** O3 (b), NO₂ (c), and BrO (d) dSCDs and the RMS (a) of the DOAS retrievals as a function of SZA for the 11 parameter configurations of the etalon correction (colors and markers as indicated by the legend).





None of the included sensitivity studies seems to perform significantly better or worse with respect to the RMS of the residual and the BrO dSCDs as a function of the SZA. Therefore, in the evaluation the results of all sensitivity runs are used and the uncertainty remaining after correcting the spectral artefacts of the etalon is then estimated from the mean and standard deviation of the ensemble of results as described in the main manuscript.



## Appendix B: Total BrO and $Br_y$ Langley plots for all sensitivity runs

**Figure B1.** BrO Langley fits for the 11 parameter configurations of the etalon correction. The mean [BrO] is 14.4 ppt. The standard deviation is 1.3 ppt and the mean fit error is 1.0 ppt.





**Figure B2.** $Br_y$ Langley fits for the 11 parameter configurations of the etalon correction. The mean $[Br_y]$ is 17.5 ppt. The standard deviation is 1.6 ppt and the mean fit error is 1.2 ppt.





## Appendix C: Mean Age of Air

Engel et al. (2002) inferred an empirical relationship between the mean age of air of a stratospheric air mass and its $N_2O$ volume mixing ratio shown in blue in the lower panel of Fig. C1. Their relationship is based on tropospheric $N_2O$ mixing

ratios around 2002, thus it needs to updated to present day $N_2O$ mixing ratios by multiplying it with the fractional increase of tropospheric $N_2O$ since 2000. This update is shown in orange in the lower panel of Fig. C1. The $N_2O$ profile of air masses probed during the balloon flight from Timmins can be inferred from the co-deployed GLORIA instrument. A mean $N_2O$ profile over the entire flight retrieved from GLORIA spectra (blue) and a $N_2O$ profile measured by MLS on August 23, 2022 over 48° N and 82.5° W (orange) (Lambert et al., 2020) are shown in the upper panel of Fig. C1. At balloon altitude we find a $[N_2O]$ of $(40 \pm 20)$ ppb resulting in a mean age of air of $(5.5 \pm 1.0)$ yrs since stratospheric entry.

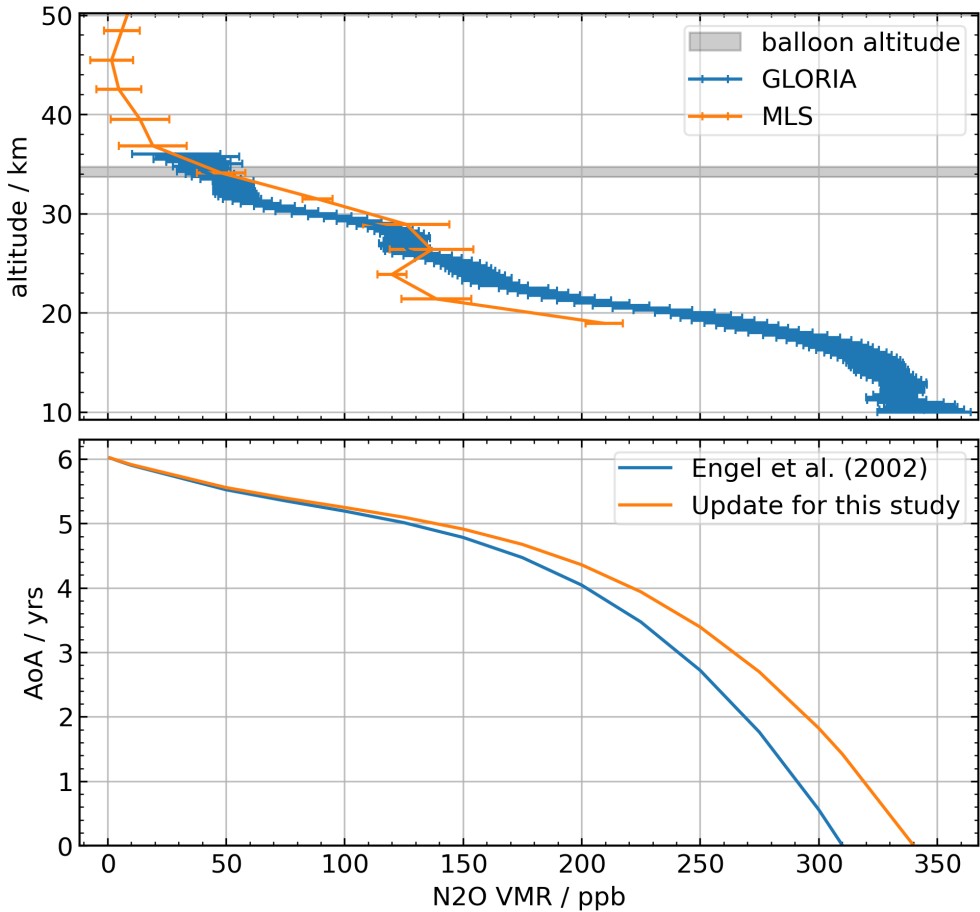

**Figure C1.** $N_2O$ profiles measured by GLORIA and MLS (upper panel). The lower panel shows the mean age of air (AoA) for a given $N_2O$ VMR. A measured $[N_2O]$ of $(40 \pm 20)$ ppb results in an age of air of around $(5.5 \pm 1.0)$ yrs since stratospheric entry.




*Author contributions.* KV, PH, and RK assembled and tested the instrument in the laboratory. KV, PH, and KP assembled the instrument on the HEMERA 1 and 2 balloon payloads and participated in the balloon campaigns and KV did the data evaluation. BF wrote the software used to operate the spectrometers during the deployments. GW, MH, and JU provided the onboard $N_2O$ data from the GLORIA instrument from which the age of air is inferred; moreover GW, MH, and BMS provided the TOMCAT model setup. KV, AB, and KP wrote the manuscript, with contributions from all co-authors. KP and AB conceived and supervised the study.


*Competing interests.* Some authors are members of the editorial board of AMT. The authors have no other competing interests to declare.

*Acknowledgements.* This work was funded by the Deutsche Forschungs Gemeinschaft, Germany (DFG) under grant numbers BU2599/9-
1 and PF384/23-1 and through the European Union (EU) project HEMERA (grant number 730970). Additional support from the Centre d'Études Spatiale (CNES), France is highly acknowledged. We are grateful to the CNES balloon team for the campaign organisation, and for successfully performing the balloon flight from Esrange/Kiruna (Sweden) on August 21, 2021 and from Timmins (Ontario, Canada) on August 23, 2022.



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
