# Peer review of "A novel, balloon-borne UV/visible spectrometer for direct sun measurements of stratospheric bromine"

_EGUsphere, 2023_

## Author Comment (AC1)

Voss et al., A novel, balloon-borne UV/visible spectrometer for direct sun measurements of stratospheric bromine

**Answers to Reviewer 1**

The reviewer's comments are in black, *answers are in blue*.

This is a well-written paper that presents the design and first results from a new balloon spectrometer for solar occultation measurements of BrO. The instrumentation is described in detail and the co-authors present improvements to the standard retrieval approach together with a convincing handling of instrumental artifacts. The inferred bromine load and its derived uncertainty shows the usefulness of these measurements for monitoring stratospheric bromine trends. The paper is a good fit for AMT and should be published after addressing these minor revisions:

1. *We thank the reviewer very much for the encouraging evaluation of our manuscript.*

The case that the 1-2 ppt difference quoted here is actually of scientific significance should be made. On the surface it seems small in light of the total budget of 20 ppt.

2. *We agree and the following sentence is added to the manuscript:*
   *This small difference in total bromine may point to some inorganic bromine being scavenged during its transport from the lower into the middle stratosphere and may accordingly reduce the bromine-mediated loss in ozone (Sinnhuber and Folkins, 2006).*

The system level descriptions of the solar tracker and the spectrometer units are well done, but would be greatly helped with schematic level drawings of the designs (in addition to the picture, which is a bit hard to follow for one not familiar with the instrument).

3. *In response to the reviewer comment the following schematic drawing of the solar tracker is added to the manuscript.*

[Figure]

*Figure 1: Schematic of the light path through the solar tracker.*

How big is the non-linearity correction in the processing step, even when the detector saturation is kept to 30-60%?

4. *We added the following sentence to the manuscript.*
   *The detector's response is most linear for pixel saturations between 30 % and 60 % with deviations from linearity within this range being smaller than 0.002 %. Thus, the correction in the processing step increases the signal of pixels with saturations between 30 % and 60 % by less than 0.002 %.*

Something is missing the explanation of how the dependence of etalon optical density on SCD is used for the retrieval process. How is the extrapolation to the larger SCDs performed throughout the retrieval range beyond the high sun range used for the characterization of the dependence?

5. *The sentence in the manuscript is changed to provide some details:*
   *The inferred principal components are then used to correct all spectra including those used for scientific analysis. To this end, the DOAS retrieval includes a set of the inferred residual spectral structures as pseudo cross sections. The scaling of each pseudo cross section together with the gas absorption structures is determined via least-squares minimization.*

The significant difference in [BrO] for different polynomial degrees is somewhat concerning. Is there some level of non-orthogonality between the higher degree polynomials and the BrO cross section? If the residuals are the same, would it not make sense to take the result for the lowest degree polynomial as the most robust?

6. *We assume that the reviewer comment refers to Figure B1 and Figure C1 of the appendix. The polynomial referred to here is not the polynomial included in the DOAS fit itself. Rather the polynomial referred to is the polynomial fitted to the optical density time series to model $\tau_{atmo}$ of each pixel within the preprocessing procedure used to infer $\tau_{residual}$. (We changed the notation from $\tau_{etalon}$ to $\tau_{residual}$ in response to the second reviewer's comments.)*

   *The polynomial represents the increasing optical density of $\tau_{atmo}$ with increasing light path through the atmosphere with decreasing solar elevation. If $\tau_{atmo}$ would only increase due to Rayleigh scattering this would imply a linear function of the total $SCD_{air}$, however also strong absorbers such as ozone and possible instrumental dependencies on the solar elevation could change the measured optical density. Both effects may not scale linearly with $SCD_{air}$. Since both effects may also be wavelength dependent, the resulting (pseudo) extinction is different for different spectral pixels. Therefore, different polynomial degrees are fitted to $\tau_{meas}$ to model $\tau_{atmo}$ for each wavelength (pixel). By that, we can infer the residual optical densities. The mean quality of the polynomial fit to $\tau_{meas}$ ($SCD_{air}$) over all pixels is found to be similar for any of the chosen polynomial degrees of order 3 to 5. However, by visual inspection, it seems that for some pixels a polynomial of $3^{rd}$ degree or lower does not capture the correct slope of $\tau_{atmo}$ ($SCD_{air}$) well, however a polynomial of $5^{th}$ or higher order may additionally compensate for some variations in $\tau_{residual}$.*

How will new CCD detectors avoid the contamination problem? It seems the cause is not well understood by the co-authors, so might it not happen again? Maybe it is from internal reflection in the focal plan array itself (this is a known phenomenon)?

7. *In past studies, both in the laboratory as well as in the field, we often used Ocean Optics QE-Pro spectrometers but did not encounter such problems of residual structures. Unfortunately, prior to the first deployment from Kiruna in 2021 a thermal incident occurred to the spectrometer unit due to an operating mistake with the current spectrometers in the laboratory. We accidentally left the Peltier detector cooling running while not providing cooling of the Peltier's warm back-side by the ice-water bath. Therefore, the spectrometer optics, electronic parts and housing heated up to 60°C while under vacuum. This resulted in outgassing of volatile chemical components within the vacuum chamber and accordingly their condensation onto the cold detector surface. For cleaning, we then reversed the process, heating the detector while cooling the lid of the vacuum chamber and evacuating the spectrometer container. By this measure, most of the condensate could be removed from the detector surface such that the residual structures were largely reduced. However, it is plausible that a residual layer of condensate remained on the detector surface causing residual optical structures with $\tau_{residual} < 10^{-3}$, which interfere with the retrieval of minor atmospheric absorbers. We are certain that what we called etalon structures is a result from this operating error and the contamination of the detector surface. Thus, we expect that by replacing the contaminated with new CCD detectors will eliminate this problem.*

   *In the revised version of the manuscript we added a short report on the thermal incident*

*to the instrument performance section of the manuscript.*
*We changed the wording "etalon structures" to "residual structures" since we realized that the initial wording is too specific and misleading for the reader.*

Prolific use of brackets for parenthetical phrases and around quantities make the text a bit difficult to read. Check with the editor for guidance on correct typesetting.

8. *The use of brackets was checked throughout the manuscript. Brackets used for parenthetical phrases were reduced by rephrasing the sentences. Square brackets around molecules are commonly used to indicate concentrations of these molecules.*

Line 78: "correct for them"

9. *The text is rephrased accordingly.*

Line 88: rephrase "all electronics threatened to overheat"

10. *The text is rephrased accordingly.*

Line 155: rephrase "were more threatened"

11. *The text is rephrased accordingly.*

*References:*

*Sinnhuber, B.-M., and I. Folkins, Estimating the contribution of bromoform to stratospheric bromine and its relation to dehydration in the tropical tropopause layer, Atmos. Chem. Phys., 6 (12), 4755–4761, doi:10.5194/acp-6-4755-2006, 2006.*

---

## Author Comment (AC2)

Voss et al., A novel, balloon-borne UV/visible spectrometer for direct sun measurements of stratospheric bromine

Answers to Reviewer 2

The reviewer's comments are in black, *answers are in blue.*

In this manuscript, the authors describe a new balloon-borne DOAS instrument for measurements of stratospheric species. The instrument's performance during two deployments is discussed, and data from one measurement flight is used to estimate stratospheric Bry and compare the results to previous estimates.

The manuscript is well written and discusses an instrument and measurements relevant to atmospheric science and the study of the ozone layer. I understand that this article is intended as a reference for future work reporting on more measurement results achieved with this instrument. I recommend it for publication after the issues listed below have been addressed.

*We thank the reviewer for the appreciation of our work and the helpful comments. Please find our point-by-point reply below.*

**Major comments**

As this is not the first stratospheric balloon-borne DOAS instrument, it would be good to highlight the actual progress of this instrument compared to the previous versions. This is not clear to the reader, and my impression is that while this instrument is new, it is not particularly innovative or better than past ones.

1. *Previously, our balloon-borne DOAS instruments were flown together with the LPMA instrument which provided the solar tracker (Ferlemann et al. 1998, Weidner et al. 2005). Since the LPMA instrument and hence the solar tracker was no longer available, a new solar tracker was needed for balloon-borne solar occultation measurements. Fortunately, in parallel, a solar tracker was developed by our group to monitor greenhouse gases from the ground on slowly moving platforms (e.g. ground-based: Gisi et al. 2011; ship-borne: Klappenbach et al. 2015, on a pickup truck: Butz et al. 2017). The concept of this ground based solar tracker was adjusted for deployment on balloon gondolas. Compared to the former LPMA solar tracker (Camy-Peyret et al. 1995), the presented stand-alone solar tracker weights much less, is more compact and consumes less power.*

   *Moreover, the concept and design of the spectrometer module was inherited from the limb spectrometers used by Weidner et al. (2005) and Kritten et al. (2010). However, in the present application the spectrometers were changed from Ocean Insight QE 2000 and QE65000 spectrometers to slightly larger, higher performance Ocean Insight QE-Pro spectrometers. Additionally, electronics including the onboard computer and power lines were updated. We expect that the new setup with better CCD detectors, in principle, offers higher sensitivity and stability than previously achieved. However, due to the spectral artefacts originating from detector contamination reported on in the revised manuscript, we*

*were not able to reduce the detection limits compared to previous balloon-borne DOAS experiments. In the future, after replacing the contaminated CCD detectors, we hope to reduce the uncertainties and to detect weaker absorbers such as IO. Compared to the LPMA/DOAS direct sun spectrometer, the new spectrometer weighs less while being more compact and thus can be placed flexibly on the gondola.*

*The presented, low weight and compact instrument consisting of both units is well suited to be deployed as secondary payload with several other instruments on the same payload.*

*The manuscript's instrument description has been updated with the information presented here.*

The description of the instrument is very detailed (I would say too detailed) in some parts, but it lacks drawings to understand the set-up of the telescope and the overall system. Figure 2 does not really help in this respect. Please add schematics providing more details on the set-up.

2. *A schematic of the solar tracker optics is added to the manuscript (see the above figure (Figure 1) added to comment #3 of reviewer 1).*

As discussed in the manuscript, there are two major problems with the current system: a) the sun-tracker is not fast enough to allow for measurements during ascent when the gondola is not stabilized, and b) there are artefacts in the spectra, which make BrO analysis difficult.

The first problem is only mentioned, and no indication is given regarding how it will be solved in future deployments. Not being able to measure profiles is a severe limitation of the instrument, so this needs to be discussed more.

3. *Yes, the reviewer is correct in his assessment.*

*The first limitation is mostly related to recent safety-related changes in the balloon flight train and modifications in the launch procedure. The operating agency (CNES) decided to replace the previously used auxiliary balloons, which were released shortly after launch, by one auxiliary balloon, that is no longer released and that has the flight train running through it. This change comes at the expense of a larger area of attack for the shear winds which induces strong torques to the flight train and the gondola. In consequence, a reliable azimuth stabilisation is not (yet) possible when strong shear winds are prevailing, i.e. during balloon ascent below altitudes of at least 10 km to 15 km. This hinders continuous ascent measurements due to the unpredictable rotations of the balloon gondola.*

*Of course, one could argue, a more rapid and improved sun-tracker, that is able to rotate freely by 360 degree without cable connections, could compensate for the deficit in the azimuth stabilisation of the gondola. However, it needs to be noted that although the sun tracker is mounted on top of the gondola its unobscured view to the sun is limited to an azimuth range of approximately 270° by the presence of the gondola structure, in particular the so-called pivot by which the gondola is azimuthally stabilized. The shading of the sun-tracker by these structures further complicates a fast and continuous tracking of the sun. The*

*realisation of such a tracker is a difficult engineering challenge which we did not pursue so far.*

*Some discussion on the above issue has been added to the manuscript.*

The second problem is discussed in great detail, but only concerning the approach taken to correct it in the spectra. Very little is said about the possible origins of the artefacts beyond the fact that they are linked to etaloning on the CCDs. I'm not entirely convinced by this explanation, as this appears to be a rapid process, and the question is, what would be condensing on the CCDs at this rate in an evacuated housing? In the sense of "what can the reader learn from this manuscript", I would hope for a more detailed discussion of possible sources and the tests performed to investigate them:

4. *As described in detail in response #7 to reviewer 1, a thermal incident due to an operating error occurred prior to the Kiruna balloon launch contaminating the surface of the CCD detector. This mal-functioning manifested itself by etalon-like residual structures in the recorded spectra which we attributed to the presence of a condensate on the surface of CCD detectors. Follow-on attempts to remove the condensate on the CCD by temperature gradient assisted vacuum cleaning were partly successful in that we could reduce the residual structures due to condensate to a degree lower than the detection limit of the residual in the laboratory. However, for the spectra recorded during the balloon flights we assume some residual condensate remained on the CCD's surface.*

   *A short report on the thermal incident is added to the instrument performance section of the manuscript.*

Has the effect been reproduced in the lab?

5. *Several lab experiments were carried out to investigate if we can reproduce the artefact in the lab after the first deployment. However, we were not able to reproduce a similar behavior on the $\tau_{residual}$ < 10$^{-3}$ level, mainly because light sources (halogen lamb, mercury and krypton emission lamps) available in laboratory were less stable than the sun. Further using the sun in test measurement at the ground led to residual structures due to variability in atmospheric processes (absorption and scattering) of the order of the instrumental effect to be studied.*

Does it occur also if only the CCDs are cooled but not the rest of the optical system?

6. *This was not directly tested, since in the lab experiments, the entire optical system was always kept at 0°C (to support the cooling of the warm side of the Peltier element) with the ice-water bath while the CCDs were usually cooled to -10°C by the thermo-electric cooler.*

Does it happen for both channels? Are the phases of the etalons on the two CCDs linked?

7.  *Yes, it happened for the UV and visible spectrometer, but it was not linked in phase and frequency (periodicity) for both spectrometers. However, the periodicities were of similar length.*

Could it be etaloning due to condensation on another surface (mirrors, fibres, filters)?

8.  *Due to the incident with the spectrometers described in response #7 to reviewer 1, it is rather likely that the artefacts are linked to the response of the spectrometers' detectors.*

Concerning the proposed correction approaches, I do not understand the rationale behind using SCD_air as the axis – I would have thought that time is the relevant parameter here. Please explain.

9.  *When describing the optical density changes in atmospheric spectra $SCD_{air}$ is the natural coordinate since Rayleigh scattering is the dominant process (e.g. Gurlit et al. 2005). And yes, very likely SCD_air is not the natural coordinate to describe oscillations of the artefacts. Therefore, we investigated the oscillations of the spectral artefacts as a function of several parameters including time, SZA and $SCD_{air}$, but the oscillations did not show a constant period as a function of any of these parameters. So, we decided to keep $SCD_{air}$ as the preferred coordinate.*

Clearly, the etaloning introduces substantial additional uncertainty even after correction, and this limits the value of the current measurements in addition to the limitation imposed by the lack of profiling capability.

10. *Yes, we concur. This is why we try to estimate the additional uncertainty by the sensitivity study. However, since the cause of this residual structure is not totally clear but did not show in previous deployments with similar spectrometers, we decided to replace the spectrometers for potential future deployments. With the replaced spectrometers, we expect the residual structures will no longer be present, permitting a lower detection limit with a lower error in the retrieved $Br_y$.*

**Minor comments**

Figure 1: What is the sharp gradient in the green shading at 80.5°? This looks like an artefact.

11. *This is an artefact of the open street map tile. In the light of open science and open data, we would like to use the map provided by open street map nevertheless.*

Figure 3: The FWHM of the UV instrument is relatively small – is a convolution of the literature cross-sections necessary (and even possible) at this high spectral resolution?

12. *Yes, it is since the literature cross sections used for $NO_2$, $O_3$ were recorded at much higher spectral resolution. The resolution of the literature $O_3$ spectra is about 0.06 nm of the literature $NO_2$ spectra is about 0.01 nm, and for the literature $BrO$ cross section it is around the same spectral resolution as our instrument but recorded in 0.04 nm intervals. For $O_4$ the*

*absorption structure is very broadband, thus the convolution just maps the absorption structure to the wavelength grid of our spectrometer.*

L264: Calling the vertical column density VD is inconsistent with the other notation. Please use VCD_air. Alternatively, if you meant density here (as suggested by your use of VD in equations 4 and 5), use another symbol such as \rho_air for air density.

*13. Air density is meant here, the notation in the manuscript is changed accordingly.*

L268: Why is it a good assumption that BrO is constant above floating altitude?

*14. A constant [BrO] within more than a scale height above balloon float altitude of 34 km is predicted by photochemical models. For example, the simulated SLIMCAT [BrO] varies by less than 5% between 34 km and 40 km altitude.*

Equation 4: This is not correct. Either you remove the delta h_i, or you replace VD with the concentration of air

*15. VD is supposed to be the density of air. For clearer notation, we changed VD to rho as suggested.*

Equation 5: Same problem as for equation 4

*16. VD is supposed to be the density of air. For clearer notation, we changed VD to rho as suggested.*

L291: predominant

*17. We changed the text accordingly.*

L461: I wouldn't say that NO2 SCDs are unaffected, but they are not much affected.

*18. We concur and changed the text accordingly.*

*References:*

*Butz, A., Bösch, H., Camy-Peyret, C., Chipperfield, M. P., Dorf, M., Dufour, G., Grunow, K., P., J., Kühl, S., Payan, S., Pepin, I., Puk,ıte, J., Rozanov, A., von Savigny, C., Sioris, C., Wagner, T., Weidner, F., and Pfeilsticker, K.: Inter-comparison of Stratospheric O3 and NO2 abundances retrieved from balloon-borne direct sun observations and Envisat/SCIAMACHY limb measurements, Atmos. Chem. Phys., 6, 1293–1314, 2006.*

Butz, A., Dinger, A. S., Bobrowski, N., Kostinek, J., Fieber, L., Fischerkeller, C., Giuffrida, G. B., Hase, F., Klappenbach, F., Kuhn, J., Lübcke, P., Tirpitz, L., and Tu, Q.: Remote sensing of volcanic CO2, HF, HCl, SO2, and BrO in the downwind plume of Mt. Etna, Atmospheric Measurement Techniques, 10, 1–14, https://doi.org/10.5194/amt-10-1-2017, 2017.

Camy-Peyret, C., Jeseck, P., Hawat, T., Durry, G., Berubeé, G., Rochette, L., and Huguenin, D.: The LPMA balloon borne FTIR spectrometer for remote sensing of the atmospheric constituents, in: 12th ESA Symposium on Rocket and Balloon Programmes and Related Research, 1995.

Ferlemann, F., Camy-Peyret, C., Fitzenberger, R., Harder, H., Hawat, T., Osterkamp, H., Schneider, M., Perner, D., Platt, U., Vradelis, P., and Pfeilsticker, K.: Stratospheric BrO profiles measured at different latitudes and seasons: Instrument description, spectral analysis and profile retrieval, Geophys. Res. Lett., 25, 3847–3850, 1998.

Gisi, M., Hase, F., Dohe, S., and Blumenstock, T.: Camtracker: a new camera controlled high precision solar tracker system for FTIR-spectrometers, Atmos. Meas. Tech., 4, 47–54, https://doi.org/10.5194/amt-4-47-2011, 2011.

Gurlit, W., Bösch, H., Bovensmann, H., Burrows, J. P., Butz, A., Camy-Peyret, C., Dorf, M., Gerilowski, K., Lindner, A., Noël, S., Platt, U., Weidner, F., and Pfeilsticker, K.: The UV-A and visible solar irradiance spectrum: inter-comparison of absolutely calibrated, spectrally medium resolution solar irradiance spectra from balloon- and satellite-borne measurements, Atmos. Chem. Phys., 5, 1879–1890, https://doi.org/10.5194/acp-5-1879-2005, 2005.

Klappenbach, F., Bertleff, M., Kostinek, J., Hase, F., Blumenstock, T., Agusti-Panareda, A., Razinger, M., and Butz, A.: Accurate mobile remote sensing of XCO2 and XCH4 latitudinal transects from aboard a research vessel, Atmos. Meas. Tech., 8, 5023–5038, https://doi.org/10.5194/amt-8-5023-2015, 2015.

Kritten, L., Butz, A., Dorf, M., Deutschmann, T., Kühl, S., Prados-Roman, C., Pukῑte, J., Rozanov, A., Schofield, R., and Pfeilsticker, K.: Balloon-borne limb measurements of the diurnal variation of UV/vis absorbing radicals – a case study on N O2 and O3, Atmos. Meas. Techn., 3, 933 – 946, 2010.

Weidner, F., Bösch, H., Bovensmann, H., Burrows, J. P., Butz, A., Camy-Peyret, C., Dorf, M., Gerilowski, K., Gurlit, W., Platt, U., von Friedeburg, C., Wagner, T., and Pfeilsticker, K.: Balloon-borne limb profiling of UV/vis skylight radiances, O3, NO2, and BrO: technical set-up and validation of the method, Atmos. Chem. Phys., 5, 1409–1422, https://doi.org/10.5194/acp-5-1409-2005, 2005.